# Standing on Elevated Platform Changes Postural Reactive Responses during Arm Movement

**DOI:** 10.3390/brainsci14101004

**Published:** 2024-10-03

**Authors:** Luis Mochizuki, Juliana Pennone, Aline Bigongiari, Renata Garrido Cosme, Marcelo Massa, Alessandro Hervaldo Nicolai Ré, Ricardo Pereira Alcântaro, Alberto Carlos Amadio

**Affiliations:** 1School of Arts, Science and Humanities, University of São Paulo, São Paulo 03828-000, Brazil; mochi@usp.br (L.M.); alinebigongiari@hotmail.com (A.B.); garrido.renatinha@gmail.com (R.G.C.); mmassa@usp.br (M.M.); alehnre@usp.br (A.H.N.R.); ricardo.junior@usp.br (R.P.A.J.); 2Department of Orthopeadics and Traumatology, Hospital das Clínicas, Faculty of Medicine, University of São Paulo, São Paulo 05402-000, Brazil; 3School of Physical Education and Sport, University of São Paulo, São Paulo 05508-060, Brazil; acamadio@usp.br

**Keywords:** posture, EMG, fear of falling

## Abstract

**Background/Objectives**: This study investigated the behavior of postural adjustments throughout the entire action: from the preparatory phase (anticipatory postural adjustment, APA), the focal movement phase (online postural adjustments, OPA), to the compensatory phase (compensatory postural adjustment, CPA) while raising the arms in a standing position, both with eyes opened and closed. The goal was to analyze the effects of reduced sensorial information and different heights on postural muscle activity during these three phases. **Methods**: Eight young women performed rapid shoulder flexion while standing on the ground and on a 1-m elevated platform. The EMG activity of the trunk and lower limb muscles was recorded during all three phases. **Results**: Although average muscle activity was similar on the ground and the elevated platform, the pattern of postural muscle activation varied across the motor action. During OPA, all postural muscle activity was the highest, while it was the lowest during APA. On the elevated platform postural muscles have increased their activation during APA. In the most stable condition (standing on the ground with eyes opened), muscle activity showed a negative correlation between APA and OPA, but there was no correlation between OPA and CPA. **Conclusions**: Our results suggest postural control adapts to sensory, motor, and cognitive conditions. Therefore, the increased demand for postural control due to the height of the support base demands greater flexibility in postural synergies and alters muscle activity.

## 1. Introduction

Anticipation is a cognitive process grounded in experience, in which predicted outcomes are crucial to selecting appropriate actions [1]. Anticipatory postural adjustment (APA) is a postural response that aims to ensure the optimal mechanical conditions for motor actions. Essentially, the APA is initiated to prevent imbalances during movement, enabling individuals to perform daily activities without falling. Belen’kii et al. (1967) proposed that muscles are activated during APA to maintain balance with minimal energy expenditure [2]. If postural muscle activity during APA fails to minimize disturbances adequately, compensatory strategies become necessary to maintain system stability [3]. Thus, APA [4,5] influences compensatory postural adjustment (CPA), which is a neuromuscular response to loss of equilibrium. The central nervous system (CNS) integrates afferent and efferent information to trigger and modulate APA and CPA [6,7]. APA can be influenced by perceptual cues [8] and by the motor action itself [9]. Aruin et al. (1998) found that high stability demand can suppress APA in an equilibrium recovery task, suggesting such a strategy may mitigate certain instability effects caused by the APA itself [10].

Perception is not a passive reception of sensory data but rather a form of probabilistic inference [11]. Unrealistic predictions can emerge from sensory misinterpretations. For instance, standing on an elevated platform may evoke a fear of falling, even when no actual physical threat exists [12,13]. Adkin et al. (2000) showed postural threat modulates postural control, indicating that psychological factors impact perception and balance. For instance, anxiety can lead to balance disturbances [14]. A person may feel less confident when standing at heights, and under such conditions, closing their eyes might further impair their ability to maintain postural stability [15]. This loss of confidence can result in avoidance behavior, such as a fear of falling. Often, fear of falling manifests as a reluctance to stand still in high places with a narrow support base or as a hesitation in older adults following an accidental fall [16,17]. Could such a misinterpretation of falling risk influence postural reactive responses? For example, the fear of falling could increase the postural responses during a focal movement more than usual. Since these responses are associated with different neural structures (the long latency component involves a transcortical loop contributing to the postural response, while the short and medium responses rely on reflex response and voluntary reactions [18,19]), could a postural threat induce different postural responses during and after the focal movement? Studies on APA and CPA often treat perturbation as a discrete focal movement. However, these studies focus on the EMG changes that occur before (APA) and after (CPA) the perturbation without addressing what happens during the execution of the focal movement. When considering the task of maintaining postural stability during and after the focal movement, three distinct mechanical demands can be observed: (1) to predict the postural perturbation and react accordingly (APA), (2) to respond to the perturbation and maintain postural stability during the movement (first reactive response), and (3) to restore postural stability after the movement, as the body configuration has changed (second reactive response). To differentiate these two reactive postural responses, we propose referring to the postural adjustments during the focal movement as online postural adjustments (OPA) and the responses occurring after the end of the focal movement as CPA. We suggest this non-realistic perception of postural threat can lead to misinterpretation and untimely altering postural responses. Moreover, these postural responses during and after the movement will differ because continuous afferent information from the focal movement and the postural set regulates the postural response in real time [20]. According to Cordo and Nashner (1982), the focal movement and postural responses are reciprocally related. Therefore, we could expect that postural responses during the focal movement and after its completion would behave differently under conditions of postural threat [20].

This study aims to analyze the effects of reduced sensorial information and different heights on postural muscle activity across these three phases. Our main hypothesis is that standing on elevated ground will induce changes in muscle activity throughout the motor action. To test this hypothesis, young women performed shoulder flexion as fast as possible while standing on the ground and on a 1-m-high portable elevated platform. Given that focal movement triggers two perturbations (when it begins and ends), the postural muscle activity during OPA may influence both the preceding APA and subsequent CPA. Our second hypothesis suggests that the postural muscle activity during the focal movement will be higher compared to APA or CPA, and this activity will be further enhanced by the elevated ground condition. Therefore, we expect postural activity during focal movement will modulate the postural muscle activity during both APA and CPA. The analysis of OPA will provide insights into how postural control responds to avoidance behavior. Additionally, we calculated the postural muscle synergies during these phases and discussed their association with the focal movement through this study.

## 2. Materials and Methods

### 2.1. Participants

Students at the University campus were invited to participate in this study. The participants were eight healthy young women (22 ± 3 years old, 1.59 ± 0.05 m tall, and 58.7 ± 4.2 kg mass). The inclusion criteria specified that participants should not have any neurological or musculoskeletal injury or disorders or balance disorders. Participants were excluded if they could not stand safely on the 1-m-high platform (0.5 m long and wide). All participants signed an informed consent form in accordance with the Ethical Committee of the School of Physical Education and Sport of the University of São Paulo.

### 2.2. Instruments

Muscle electrical activity (EMG) and the shoulder angle were measured with a data sampling frequency of 1 kHz. A two-dimensional flexible electrogoniometer (NorAngle II, Noraxon, Scottsdale, AZ, USA) was used to assess shoulder motion. EMG was recorded using an 8-channel EMG system (Myosystem 1400, Noraxon, Scottsdale, AZ, USA) with a band-pass filter set between 20 and 500 Hz, input impedance greater than 10 MΩ, common mode rejection ratio exceeding 85 dB, noise Ratio below one μV RMS, and a total amplification factor of 1000. Active (differential and pre-amplified) surface electrodes were employed to measure the following muscles: anterior deltoid (AD), lumbar extensor (LE), rectus abdominis (RA), rectus femoris (RF), biceps femoris (BF), tibialis anterior (TA), and gastrocnemius lateralis (GL). Electrode placement adhered to SENIAN procedures [21], maintaining a one cm distance between electrode centers. Both measurement systems were connected to a data acquisition system controlled by Myosystem 1400, Noraxon, Scottsdale, AZ, USA software installed on a PC Pentium 4 2.66 MHz computer.

### 2.3. Motor Task

In a standing position (feet parallel, ankles and haluxes touching), participants were instructed to raise both arms as quickly as possible, with elbows extended by flexing the shoulders while holding a 2-kg dumbbell in each hand. The movement stopped when both arms were parallel to the ground. The initial position involved standing quietly with arms relaxed at the sides of the trunk. The final position required standing with arms extended in front of the body, parallel to the ground. Participants should hold the final position for about three seconds before returning to the initial position.

This motor task was repeated ten times for each condition: eyes open and eyes closed on both the ground and a 1-m-high platform (0.5 m long and wide). When their eyes were opened, participants should look at a 0.5 m radius circular target on a wall 2.5 m ahead. The task was self-initiated, with each participant completing four sets of ten repetitions (one set in each condition). To minimize reaction time effects, participants were allowed to begin the task whenever they wished. A three-minute rest interval was implemented between each set to avoid fatigue. The order of conditions was randomly assigned.

### 2.4. Variables and Data Analysis

The raw EMG signal had the mean removed, full-wave rectified, and low-pass filtered using a 4th-order Butterworth filter with a 200 Hz cutoff frequency. The shoulder joint angle signal was similarly low-pass filtered with a 4th-order Butterworth filter, using a cutoff frequency of 20 Hz. For each trial, the processed EMG and shoulder angle data were segmented into three epochs: APA, OPA, and CPA. The initial t_0_ and final t_f_ points of shoulder movement were defined by the onset of the angular acceleration, obtained by double derivation of the angular position measured by the electrogoniometer.

The epoch limits were defined as follows: APA (from 200 ms before t_0_ to 50 ms after t_0_), OPA (from 50 ms after t_0_ to t_f_), and CPA (from t_f_ to 250 after t_f_). The limits for the APA phase (t_1_ and t_2_) vary in the literature, with t_1_ typically ranging from -200 ms to -100 ms and t_2_ from 0 ms to 50 ms [22,23,24]. For our study, we selected the limits t_1_ and t_2_ to maximize the epoch length while preserving data integrity.

The processed EMG signals for each postural adjustment were transformed into EMG principal components (EMG-PC) using the principal component analysis (PCA). PCA is a linear transformation technique used to compute eigenvectors and eigenvalues of a data matrix [25]. We applied PCA to calculate two, three, or four EMG-PCs from the same set of processed EMG signals. This variation was utilized to assess the effects of dimensionality reduction within muscle synergy.

For independent synergies, the EMG-PC was analyzed using independent component analysis (ICA) [26]. The independent components (IC) are statistically independent; therefore, we refer to the EMG-IC as muscle synergies [27]. All data processing and time series analysis were conducted using Matlab (version 2009b; Mathworks, Natick, MA, USA) scripts.

From the processed EMG of each muscle, the root mean square (RMS) value was calculated for the APA, OPA, and APC epochs. The following parameters were derived from these epochs. To distinguish these parameters, we utilized Winter’s proposal [28] regarding the levels of organization in neuro-musculoskeletal-joint integration for movement execution.

The parameters of the basic mechanisms of neuromuscular regulation included: EMG intensity:The RMS value normalized by 95% of the maximum RMS value measured during the repetition.Muscle latency: the time between the onset of muscular activation and the beginning of the focal movement. The latency of each muscle was calculated as the time for the first EMG signal prior to the focal movement, which was three times greater than the baseline signal plus three standard deviations of the EMG measured during the pre-activity phase.R index (Equation (1)): the absolute value of the difference between the EMG intensity of the agonist and antagonist muscle of the respective joint. This parameter reflects the level of reciprocal inhibition [29].
(1)$$R\;=\left|\int EMG_{agonist}-\int EMG_{antagonist}\right|$$

4.C index (Equation (2)): module of the sum of the EMG intensity of the agonist plus antagonist muscle of the referred joint. This parameter represents the coactivation parameter.


(2)
$$C\;=\left|\int EMG_{agonist}+\int EMG_{antagonist}\right|$$


Indicator of the postural synergistic action:5.Synergy: set of an independent component from the processed EMG (EMG-ICn) [27].6.Synergy variability—relative participation of each PC in the total variance of the EMG-PCn.

### 2.5. Statistical Analysis

For the statistical analysis, the variance analysis (ANOVA) was applied, assuming a significance level of 5%. Different comparisons were conducted: (a) RMS of each muscle across muscles, conditions (height of support base and vision), and types of postural adjustments; (b) muscle latency across muscles and conditions; (c) R and C indexes across joints, conditions, and types of postural adjustments; (d) the accounted variability of the first three PC across conditions and postural adjustments; and (e) the minimum number of principal components required to achieve at least 75% of the total variance across the conditions and postural adjustments. A Tukey HSD post hoc test was used to identify differences between the factors. For multiple linear regression, we compared the RMS of residuals across the dimensions and postural adjustments. Statistical analysis was performed using Statistica (version 5.1, 1996, USA).

## 3. Results

Muscle activity (Table 1) was not influenced by the height of the support base. However, the activity of all muscles was affected by the type of postural adjustment (F_2,1035_ > 63, *p* < 0.0001). The post hoc Tukey HSD test indicated the highest activity of all muscles occurred during the OPA phase, while the lowest activity was observed during the APA (except for the AD) or the CPA (*p* < 0.0001). The interaction between postural adjustment and support base height affected the activity of DA, RA, RF, and GL muscles (F_2,1035_ > 4.7, *p* < 0.008). The post hoc test revealed the AD muscle exhibited the greatest activity during the APA phase and on the ground (except for OPA); the RA was the most active on the ground across all postural adjustments; the RF muscle showed the highest activity at the elevated platform; and the GL muscle also had the highest activity at the elevated platform across all Postural Adjustments (*p* < 0.001). Additionally, the BF muscle activity was affected by visual information (F_1,1035_ = 3.9, *p* < 0.05), with the highest activity occurring when the task was performed with the eyes closed (*p* = 0.02).

Correlation analysis was conducted to assess the relationships between muscle activities in the APA and OPA phases, the OPA and CPA phases, and the APA and CPA phases. For the comparison between APA and OPA phases, muscle activities were correlated across all conditions: ground with eyes opened (R^2^ = 0.64 *p* = 0.02), ground with eyes closed (R^2^ = 0.63 *p* = 0.02), and elevated with eyes closed (R^2^ = 0.88 *p* < 0.001). However, when participants were on the elevated platform with their eyes open, the correlation was weak (R^2^ = 0.18 *p* = 0.19).

In contrast, muscle activities between APA and CPA showed no significant correlations across all conditions: ground with eyes open (R^2^ = 0.11 *p* = 0.54), ground with eyes closed (R^2^ = 0.15 *p* = 0.65), elevated with eyes open (R^2^ = 0.05 *p* = 0.43), and elevated with eyes closed (R^2^ = 0.02 *p* = 0.40). For the OPA and CPA comparison, muscle activities were also not correlated across most conditions: ground with eyes open (R^2^ = 0.21 *p* = 0.17), ground with eyes closed (R^2^ = 0.10 *p* = 0.54), and elevated with eyes closed (R^2^ = 0.22 *p* = 0.16). However, when participants were on the elevated platform with their eyes open, a significant correlation was observed (R^2^ = 0.65 *p* = 0.02).

The R index (Figure 1) was influenced by the height of the support base (F_1,1025_ = 22.6, *p* < 0.0001), type of postural adjustment (F_2,1035_ = 92.7, *p* < 0.0001), and joints (F_2,2070_ = 92.7, *p* < 0.0001). The post hoc Tukey HSD test indicated that the R index was the highest on the ground during the APA phase and at the ankle (*p* < 0.0001), while it was the lowest during the OPA phase and at the knee (*p* < 0.0001).

The C index (Figure 2) was influenced by postural adjustment (F_2,1035_ = 3313, *p* < 0.0001) and joints (F_2,2070_ = 339, *p* < 0.0001). The Tukey HSD post hoc test revealed that the C index was the highest during the OPA phase (*p* < 0.0001) and at the knee (*p* < 0.0001), while it was the lowest during the APA phase and at the ankle (*p* < 0.0001).

The first three principal components variances related to postural synergies (Figure 3) were affected by the postural adjustment (F_2,1035_ > 473, *p* < 0.001), and two of them (PC1 and PC3) were affected by the support base height (F_1,1035_ > 4.4, *p* < 0.04). The post hoc Tukey HSD test showed the PC1 explained variance was the highest in APA and at the ground and the lowest in OPA (*p* < 0.0001); the PC2 explained variance was the highest in OPA and the lowest in APA (*p* < 0.0001); and PC3 explained variance was the highest in OPA and at the top and the lowest in APA (*p* < 0.0001).

The comparison of muscle latency (ms) across all conditions is shown in Figure 4. There were no significant differences in muscle latency when standing on a 1-m-high portable elevated platform or with eyes open versus closed.

## 4. Discussion

In this study, young women performed shoulder flexion as fast as possible while standing on both the ground and a 1-m-high portable elevated platform. We recorded the EMG of the trunk and lower limb muscles during the preparatory, execution, and compensatory postural phases. While the average muscle activity was similar on the ground and the elevated platform, the activation patterns of postural muscles varied across the motor action. All postural muscles exhibited the highest activity during OPA and the lowest in APA. Standing on an elevated platform altered the activation of the trunk and lower limb muscles during the upper limb task: the RA muscle showed reduced activation, whereas the RF and GL muscles demonstrated increased activation. In the most stable condition (on the ground with eyes open), muscle activity was negatively correlated between APA and OPA, with no correlation found between OPA and CPA phases. These results indicate the adaptability of postural activity to sensory constraints and perceived threat. Notably, despite the absence of mechanical differences, standing on the elevated platform increased muscle activity during OPA, while muscle activity during APA decreased. This behavior suggests that the elevated condition is really perceived as a postural threat.

The timeline of muscle activation illustrated the dynamics of postural activities throughout the motor action. Our results indicate that muscle activity varies across the reactive postural control phases [20], supporting the division of postural muscle activity into three distinct phases. Traditionally, postural studies often consider only one compensatory phase [5,23,30,31,32,33,34], with CPA beginning at the execution of the FM. In our framework, we defined the OPA phase as the activity of postural muscles occurring during the execution of the FM, while CPA is defined as the activity that starts at the conclusion of the FM and continues for 250 into the follow-through phase. During OPA, rapid corrections are based on reafference control, whereas feedback control during CPA addresses any postural issues related to balance body orientation or instability resulting from the follow-through. Other studies have also proposed different phases for anticipatory postural adjustments [22] and reactive control. Latash and his colleagues introduced the concepts of early anticipatory postural adjustments and early anticipatory synergy adjustments [22] as critical mechanisms in postural control.

Reduced confidence due to postural threat alters the relationships between APA and OPA. While muscle activity during the focal movement and anticipatory postural activity were correlated, this association diminished with eyes open. The postural threat posed by the elevated platform is influenced by visual information. On the other hand, standing on the elevated platform with eyes open led to muscle activity being associated with OPA and CPA. This postural threat creates different associations between muscle activations. On the ground, the focal movement triggers the preparation phase (during APA) but does not modulate postural activity during the compensatory phase. However, when participants were on the elevated platform, a lack of confidence became apparent with their eyes closed, resulting in altered relationships. Although the focal movement did not trigger muscle activation during the preparatory phase with their eyes open, it did so during the compensatory phase. Perception may have affected these relationships as the conditions of the task remained unchanged. This finding supports the hypothesis that the postural threat induced by the elevated platform leads to changes in muscle activity throughout the entire motor action.

The elevated platform, which induces a postural threat, has altered the organization of postural responses and motor action. When comparing high and low support conditions, activation of the RA decreased, while the RF and GL exhibited increased activation on the elevated platform. This suggests a higher postural demand during the task for these biarticular muscles. Adkin et al. (2002) and Adkin et al. (2003) demonstrate that APA activity is inversely associated with the fear of falling [12,35]. Their findings indicated that participants performing shoulder flexion on an elevated support base or near the edge of the support base displayed less anticipatory activity. Similarly, Gendre et al. (2016) found that the effects of fear of falling depend on initial environmental conditions and the direction of APA in relation to the postural threat [36]. However, other studies examining different tasks, such as gait [13], rise-to-toes [37,38], and handle pulls [39], also on elevated platform, reported varying effects of threat on posture. These studies observed greater postural sway [13,37,38], increased APA amplitude [38] and delayed APA generation [13] highlighting the complexity of how postural threats impact motor control.

The mechanisms by which fear of falling led to alterations in APA remain unclear. However, it appears that the CNS attempts to regulate this perceived threat using strategies that vary depending on the task and correlate with the intensity of the perceived fear.

Agonist and antagonist muscle activations exhibit distinct patterns across joints and postural adjustments. The indices of coactivation and reciprocal inhibition varied significantly depending on the joints and type of postural adjustments. This modulation of postural control, influenced by muscle pairs, can be understood through changes in these two indices [5,8]. Notably, reciprocal inhibition decreased on the elevated platform, while coactivation remained consistent, highlighting the impact of both mechanical and perceptual factors on balance control. Although the mechanical conditions remained unchanged with the elevated platform, the increased coactivation during OPA suggests a heightened response to the fear of falling [40]. Our results indicate that greater muscle activity in certain postural and focal muscles on the elevated support base reflects an increase in voluntary effort to execute the same motor task.

The highest level of reciprocal inhibition was observed during APA at the ankle, particularly when the number of dimensions was minimized. In contrast, the greatest number of dimensions emerged during OPA, accompanied by increased coactivation, especially at the knee, when participants performed the task on the elevated support base. These findings suggest that the organization of postural adjustments is highly adaptable. Matrix factorization methods are employed to identify muscle synergies [41] or muscle modes [22], which reflect how muscle groups work together based on the demands of the task. Our results suggest that antagonistic muscles exhibited reduced activation to facilitate movement in simpler tasks, while they engaged more intensively and prevented movement in more complex tasks, such as those performed at elevated support bases. This highlights the intricate interplay between muscle coordination and task complexity in postural control.

Muscle latency remained unchanged when participants stood on the 1-m-high portable elevated platform, indicating that the perceived postural threat was not mechanical in nature, as the elevated base was completely fixed and stable. When comparing the principal components from tasks performed on the ground versus the elevated platform, we observe a decrease in the variability accounted for by PC1 and an increase in PC3. The greater contribution of PC1 during ground tasks highlights a more conservative postural control, while the increased involvement of PC3 on the elevated platform suggests a more restorative role aimed at maintaining postural stability. This shift indicates that while postural control on the ground utilizes fewer dimensions, the execution of focal movements on the elevated platform requires the engagement of more principal components, reflecting an increase in the complexity of control mechanisms employed.

This study has limitations that should be considered. First, the small sample size may limit statistical power and generalizability. Although existing research suggests no significant differences in the neuromuscular system between sexes, our exclusively female sample may not adequately represent the broader population. Lastly, while we observed changes in muscle activity when participants performed movements on the ground versus the elevated platform, we did not quantify the fear of falling using a perception scale. This absence of measurement mirrors limitations in other related studies [13,37,39], making it difficult to attribute changes in muscle activity to fear definitively.

## 5. Conclusions

This study investigated how postural control adapts to different sensory and motor conditions during fast shoulder flexion movements on an elevated platform. Our results indicate that muscle synergies and muscle activity vary significantly across the postural adjustments, particularly under postural threat conditions, such as standing on an elevated platform.

Specifically, we observed that (1) muscle activity was highest during the OPA, suggesting greater postural adjustment demands during movement execution; (2) the increased height of the platform led to a reduction in RA activity and an increase in RF and GL activities, indicating a redistribution of postural control demands; (3) muscle activity in the APA was negatively correlated with OPA in stable conditions (on the ground, eyes open), but this correlation disappeared under threat conditions (elevated platform), showing that postural control is affected by perceptions of risk; and (4) the perceived postural threat on the elevated platform resulted in higher muscle activation during OPA, even though there were no mechanical differences in the environment.

These findings suggest that postural control is highly adaptable and modulated in response to sensory, motor, and cognitive changes, with increased flexibility in muscle synergies required in more challenging postural environments.

## Figures and Tables

**Figure 1 brainsci-14-01004-f001:**
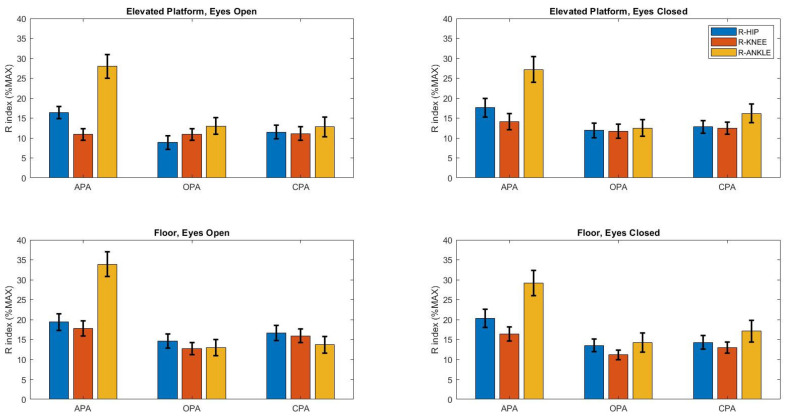
Mean ± standard deviation of R index (%MAXIMUM) according to the height of the support base (floor and elevated platform), vision (open and closed), and postural adjustment conditions. Anticipatory postural adjustment, APA; online postural adjustment, OPA; compensatory postural adjustment, CPA. Floor indicates that the R index was higher when participants performed the task on the floor than on the elevated platform (*p* < 0.0001). APA indicates that the R index was higher during APA than during OPA or CPA. R-ankle indicates that the R index was higher for the ankle joint than for the hip and knee joints (*p* < 0.0001).

**Figure 2 brainsci-14-01004-f002:**
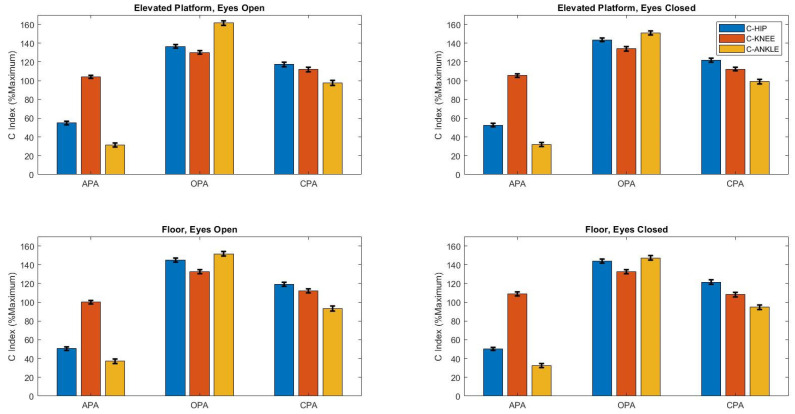
Mean ± standard deviation of C index (%MAXIMUM) according to the height of the support base (floor and elevated platform), vision (open and closed), and postural adjustments conditions. Anticipatory postural adjustment, APA; online postural adjustment, OPA; compensatory postural adjustment, CPA). OPA indicates that the C index was higher during OPA than during APA or CPA (*p* < 0.0001). R-knee indicates that the R index was higher for the knee joint than for the hip and ankle joints (*p* < 0.0001). APA” indicates that the R index was lower during APA than during OPA or CPA (*p* < 0.0001). R-ankle” indicates that the R index was lower for the ankle joint than for the hip and knee joints (*p* < 0.0001).

**Figure 3 brainsci-14-01004-f003:**
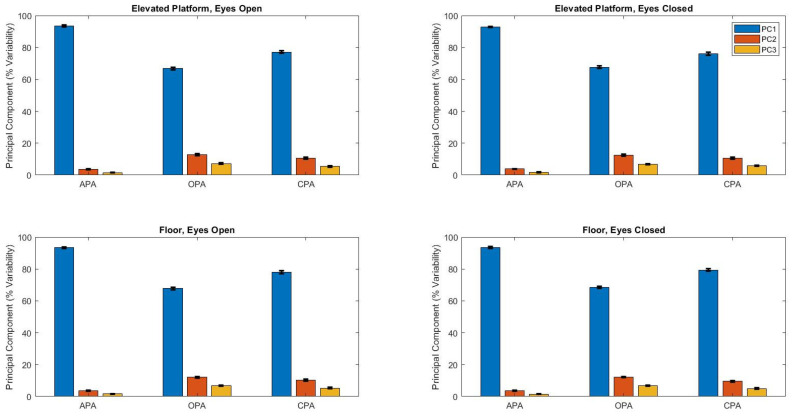
Mean ± standard deviation of the principal component accounted for variability according to the height of the support base (floor and elevated platform), vision (open and closed), and postural adjustments. Principal component 1, PC1; Principal component 2, PC2; Principal component 3, PC3; anticipatory postural adjustment, APA; online postural adjustment, OPA; compensatory postural adjustment, CPA.

**Figure 4 brainsci-14-01004-f004:**
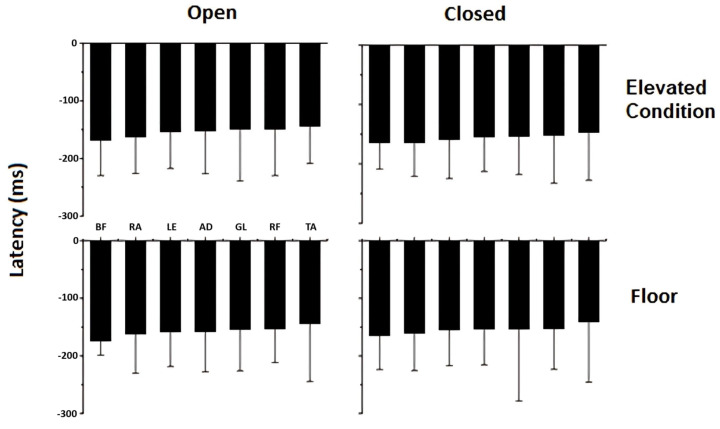
Mean ± standard deviation (n = 80) of latency for the following muscles: anterior deltoid (AD), lumbar extensor (LE), rectus abdominis (RA), rectus femoris (RF), biceps femoris (BF), tibialis anterior (TA), and gastrocnemius lateralis (GL) according to the height of the support base (floor and elevated platform) and vision conditions (open and closed).

**Table 1 brainsci-14-01004-t001:** Mean ± standard deviation of muscle activity amplitude (%MAXIMUM) according to the height of the support base (floor and elevated platform), vision (open and closed), and postural adjustment conditions.

Height	Vision	Postural Adjustment	AD	LE	RA	RF	BF	TA	GL
**Floor**	**Open eyes**	**APA (n = 80)**	56.0 ± 21.8	35.0 ± 10.8	15.6 ± 6.6	47.7 ± 11.4	52.5 ± 16.7	1.7 ± 1.2	35.5 ± 29.7
		**OPA * (n = 80)**	66.4 ± 18.1	73.6 ± 15.6	71.4 ± 8.3	65.9 ± 8.1	66.8 ± 11.6	78.3 ± 9.8	73.4 ± 12.6
		**CPA (n = 80)**	48.3 ± 15.4	57.2 ± 15.5	62.1 ± 10.1	55.5 ± 10.6	56.7 ± 16.7	47.0 ± 10.9	46.4 ± 10.7
	**Closed eyes**	**APA (n = 80)**	63.6 ± 31.3	35.1 ± 11.0	15.2 ± 8.0	54.4 ± 45.8	54.6 ± 14.9	1.7 ± 1.1	30.9 ± 27.4
		**OPA * (n = 80)**	66.7 ± 19.3	73.6 ± 15.6	70.2 ± 7.4	67.3 ± 10.3	65.0 ± 11.7	76.4 ± 11.0	70.8 ± 11.0
		**CPA * (n = 80)**	46.3 ± 16.7	59.2 ± 16.6	62.3 ± 8.8	54.8 ± 11.2	53.5 ± 14.3	46.5 ± 12.6	48.2 ± 12.0
**Elevated platform**	**Open eyes**	**APA (n = 80)**	51.2 ± 18.0	35.2 ± 9.3	19.8 ± 9.5	49.9 ± 11.5	54.1 ± 15.4	1.8 ± 1.2	29.8 ± 25.2
		**OPA * (n = 80)**	65.3 ± 15.7	66.5 ± 8.8	70.1 ± 9.4	63.5 ± 8.5	66.6 ± 10.8	76.4 ± 10.8	84.9 ± 21.1
		**CPA (n = 80)**	52.5 ± 13.5	57.8 ± 12.3	59.5 ± 9.3	55.6 ± 11.1	56.3 ± 14.2	50.3 ± 12.4	47.4 ± 10.1
	**Closed eyes**	**APA (n = 80)**	52.4 ± 28.2	35.0 ± 8.7	17.8 ± 7.6	51.3 ± 12.3	54.2 ± 16.6	2.4 ± 2.3	29.4 ± 25.4
		**OPA * (n = 80)**	67.2 ± 14.6	71.2 ± 12.2	72.3 ± 8.5	67.2 ± 15.6	66.9 ± 12.5	77.9 ± 10.5	73.2 ± 13.4
		**CPA (n = 80)**	52.1 ± 17.0	60.1 ± 15.0	61.7 ± 8.2	57.1 ± 11.1	55.2 ± 13.4	47.9 ± 12.0	51.1 ± 10.9

Anterior deltoid AD, lumbar extensor LE, rectus abdominis RA, rectus femoris RF, biceps femoris BF, tibialis anterior TA, gastrocnemius lateralis GL, anticipatory postural adjustment, APA; online postural adjustment, OPA; compensatory postural adjustment, CPA). OPA * indicates that muscle activity was the highest during OPA compared with APA and CPA (*p* < 0.0001).

## Data Availability

The raw data supporting the conclusions of this article will be made available by the authors on request.

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
