# Peer review of "Standing on Elevated Platform Changes Postural Reactive Responses during Arm Movement"

_brainsci, 2024, doi:10.3390/brainsci14101004_

Round 1
Reviewer 1 Report
Comments and Suggestions for Authors
see attached.

Author Response
We are thankful for all the comments Reviewer 1 did. We are sure the manuscript has improved its quality
Comment 1: The reviewer asked for more explanations for these statements: "Predictions are based on perceptions. Non-realistic predictions can emerge from wrong assumptions built from misinterpretations."
Response: We have rewritten these phrases.
Comment 2: Justify the sample size.
Response: We understand the concerns about a small sample size. But, we also believe it is important to consider that such a small sample increases homogeneity, providing relevant and reliable insights, and allowing for a more focused analysis. This is an exploratory study, which allowed us to propose an idea about postural control. The nature of the experimental setup, which involves standing on a 1 m high and 0.5 m wide platform, may introduce physical limitations for recruiting a large sample. This issue was also discussed as a limitation.
Comment 3: The authors are recommended to summarize the main findings of this study- not to replicate data which are already listed in tables.
Response: We have rewritten to The Results section to make it as direct and short as possible without losing any necessary detail.
Comment 4: typo. please fix it through the text.
Response: We have fixed this typo error.
Reviewer 2 Report
Comments and Suggestions for Authors
This work explores the effect of motor, sensory and cognitive context to the activity of muscles during postural control. The study idea is interesting and the methodology sounds. However, I have few comments about the clarity of the overall manuscript.
Major Comments
Both the introduction and the discussion of the manuscript includes a large number of references, many of them revolving around the topic of discussion but not strongly focused on the specific study presented which makes difficult to understand what the authors are trying to elucidate. The manuscript also shows an important number of typos that further difficult following the workflow of the explanations (few detailed in 'Minor Comments').
The use of tables to present most of the results makes also difficult to visualize and contextualize the changes in muscle activity reported by the authors. The use of bar graphs combined with a smart representation of the statistical differences reported will allow authors to emphasize in a clear way the difference of muscle activity among motion phases and task conditions.
Authors perform a synergy analysis, but they do not show the traditional representation of the synergies in terms of temporal and spatial components. I think this kind of representation would be very useful for readers to appreciate inter-condition differences and also might allow the authors to discuss about differences in the motor control strategy from the point of the CNS.
Conclusions are too generic. Authors just state that sensory, motor and cognition conditions affect muscle synergies and therefore muscle activity changes. If these specific changes were more clearly stated and represented during the results and discussion section it would be possible to make a quick reference to them in the conclusions.
Minor Comments
Line 16 --> "... as-fastas-they-could ... " should be "... as-fast-as-they-could ..."
Line 52 --> there is a parenthesis that opens and never closes.
Line 78 --> there a letter "e" in the middle of the sentence
Line 96 --> "Our second hypothesis the postural muscles..." --> maybe authors wanted to write: "Out second hypothesis is that the postural muscles..."
Comments on the Quality of English LanguageOverall English must be reviewed
Author Response
We thank the Reviewer 2 for all comments, we expect to have improved the quality of our manuscript after the Review Process. These are our responses to each comment.
Major Comments
Both the introduction and the discussion of the manuscript includes a large number of references, many of them revolving around the topic of discussion but not strongly focused on the specific study presented which makes difficult to understand what the authors are trying to elucidate.
Response: We have rewritten the Introduction and Discussion sections.
The manuscript also shows an important number of typos that further difficult following the workflow of the explanations (few detailed in 'Minor Comments').
The use of tables to present most of the results makes also difficult to visualize and contextualize the changes in muscle activity reported by the authors. The use of bar graphs combined with a smart representation of the statistical differences reported will allow authors to emphasize in a clear way the difference of muscle activity among motion phases and task conditions.
Response: We have changed the tables for figures to present our data.
Authors perform a synergy analysis, but they do not show the traditional representation of the synergies in terms of temporal and spatial components. I think this kind of representation would be very useful for readers to appreciate inter-condition differences and also might allow the authors to discuss about differences in the motor control strategy from the point of the CNS.
Response: Traditional representation of EMG data was applied when data was described for each postural adjustment.
Conclusions are too generic. Authors just state that sensory, motor and cognition conditions affect muscle synergies and therefore muscle activity changes. If these specific changes were more clearly stated and represented during the results and discussion section it would be possible to make a quick reference to them in the conclusions.
Response: We have rewritten the Conclusion.
Minor Comments
Line 16 --> "... as-fastas-they-could ... " should be "... as-fast-as-they-could ..."
Line 52 --> there is a parenthesis that opens and never closes.
Line 78 --> there a letter "e" in the middle of the sentence
Line 96 --> "Our second hypothesis the postural muscles..." --> maybe authors wanted to write: "Out second hypothesis is that the postural muscles..."
Response: We have corrected these issues.
Reviewer 3 Report
Comments and Suggestions for Authors
My major concern is the newly introduced term “operational postural adjustment” (OPA). APA and PCA refer to solid concepts and long period of use in experimental and theoretical physiology. As for OPA, frankly speaking, I encounter such term for the first time. I want to be sure that this term has some solid background. If this term is coined and introduced by authors, they must present it in a much more clear manner. I would like to see difference between APA, OPA and CPA, as it is seen between APA and CPA. To my knowledge, APA (or preprogrammed corrections) (Latash, 1996) are started in advance actual movement by 60-80 ms, while corrective postural reactions are triggered in response to loss of balance (following 100 ms after the onset of movement). In addition, corrective voluntary reactions (500 ms after the onset of movement) can take place (Horak, Mancini, 2013, Latash, 1996).
Thus, APA refers to proactive postural mechanisms, while PCA – to reactive mechanisms. I can’t see place for OPA in this paradigm. To be sure that OPA is an actual phenomenon I would like to watch an EMG record with descrete APA, OPA, and PCA. It may be well so that OPA represents just continuation of APA, which started some milliseconds earlier.
I cannot provide my final judge on the manuscript before addressing this issue.
What is the functionality of OPA?
For reference
Horak F.B., Mancini M. Objective biomarkers of balance and gait for Parkinson's disease using body-worn sensors // Mov. Disord. 2013. Vol. 28, â„– 11. P. 1544–1551. doi: 10.1002/mds.25684.
There are miswriting scattered through the text (for example, line 338 – them should be than?). Please, check English writing and grammar.
Comments on the Quality of English LanguageThere are some miswritings throughout the text.
Author Response
We thank the Reviewer 3 for all provided comments. We will address each one in the following.
My major concern is the newly introduced term “operational postural adjustment” (OPA). APA and PCA refer to solid concepts and long period of use in experimental and theoretical physiology. As for OPA, frankly speaking, I encounter such term for the first time. I want to be sure that this term has some solid background. If this term is coined and introduced by authors, they must present it in a much more clear manner. I would like to see difference between APA, OPA and CPA, as it is seen between APA and CPA. To my knowledge, APA (or preprogrammed corrections) (Latash, 1996) are started in advance actual movement by 60-80 ms, while corrective postural reactions are triggered in response to loss of balance (following 100 ms after the onset of movement). In addition, corrective voluntary reactions (500 ms after the onset of movement) can take place (Horak, Mancini, 2013, Latash, 1996).
Thus, APA refers to proactive postural mechanisms, while PCA – to reactive mechanisms. I can’t see place for OPA in this paradigm. To be sure that OPA is an actual phenomenon I would like to watch an EMG record with descrete APA, OPA, and PCA. It may be well so that OPA represents just continuation of APA, which started some milliseconds earlier.
I cannot provide my final judge on the manuscript before addressing this issue.
What is the functionality of OPA?
Response: We agree with Reviewer 3 about some aspects of OPA. OPA is included in the CPA. But, classic CPA could be split into OPA and this new version of CPA because no other studies have considered the focal movement a follow-through action. Thus, thinking as a follow-through action, we were able to show some interesting insights about postural control.
There are miswriting scattered through the text (for example, line 338 – them should be than?). Please, check English writing and grammar.
Responses: We have corrected these errors.
Round 2
Reviewer 2 Report
Comments and Suggestions for Authors
I would recommend this work for publication in this current form.
Author Response
We thank Reviewer 2 for the important comments that allowed us to improve our manuscript.
Reviewer 3 Report
Comments and Suggestions for Authors
Comment:
I would like to know rationale behind equations for computation of C (co-activation index) and R (reciprocal inhibition index). There are alternative equations, for example in the study by Ervilla et al. [2012] and many other studies:
Ervilha UF, Graven-Nielsen T, Duarte M. A simple test of muscle coactivation estimation using electromyography. Braz J Med Biol Res. 2012;45(10):977-81. doi: 10.1590/s0100-879x2012007500092.
Please, provide a reference to support the applied equations.
The rationale for introduction of the OPA concept so far looks weak. EMG record is needed to represent OPA. If OPA is considered as a part of CPA, then how they are distinguished?
Author Response
Thank again for the comments the Reviewer 3 did about our manuscript. We will answer each comment in the following paragraphs.
Comment 1 - I would like to know rationale behind equations for computation of C (co-activation index) and R (reciprocal inhibition index). There are alternative equations, for example in the study by Ervilla et al. [2012] and many other studies:
Ervilha UF, Graven-Nielsen T, Duarte M. A simple test of muscle coactivation estimation using electromyography. Braz J Med Biol Res. 2012;45(10):977-81. doi: 10.1590/s0100-879x2012007500092.
Please, provide a reference to support the applied equations.
Response: In fact, there are different ways to calculate the co-activation index, as Ervilha et al. 2012 has shown. We are using the approach that Latash (for example, Slijper and Latash 2004) uses to calculate the R and C indexes. We are going to add this reference to the manuscript.
Slijper H, Latash ML. The effects of muscle vibration on anticipatory postural adjustments. Brain Res. 2004 Jul 23;1015(1-2):57-72. doi: 10.1016/j.brainres.2004.04.054. PMID: 15223367.
Comment 2: The rationale for introduction of the OPA concept so far looks weak. EMG record is needed to represent OPA. If OPA is considered as a part of CPA, then how they are distinguished?
Response. Thank you for this observation. We have changed the manuscript to improve this rationale. In APA-CPA studies, the perturbation is usually a discrete focal movement. But, those studies address no considerations about what happens during the focal movement and after the end of it, because such experimental protocols are only concerned about the perturbation induced by the beginning of the focal movement. Even the aim of the focal movement changes, such studies are concerned about how EMG changes before (APA) and after (CPA) the perturbation. But, if we consider the task is to maintain postural stability during and after the focal movement, it is possible to observe three different mechanical demands: 1) to predict the postural perturbation and do something about it (APA); 2) to respond to the postural perturbation and maintain the postural stability during the movement (CPA1); 3) to maintain the postural stability after the end of the focal movement because the body configuration is not the same, compared to before the begging of the focal movement (CPA2). We preferred to refer to CPA1 as OPA because it happens while focal movement is on and does not have a fixed epoch duration; while we are CPA2 as CPA.
Round 3
Reviewer 3 Report
Comments and Suggestions for Authors
Still, I think that some sentences can be added in support of the idea of OPA, and present its distinction from APA anf PCA.
Author Response
This discussion with Reviewer 3 is assisting us to improve our manuscript. We thank for that.
Comment 1. Still, I think that some sentences can be added in support of the idea of OPA, and present its distinction from APA anf PCA.
Response: We have added these sentences:
Moreover, these postural responses during and after the movement will differ because continuous afferent information from the focal movement and the postural set regulates the postural response in real-time (Cordo and Nashner, 1982). According to Cordo and Nashner (1982), there is a reciprocal relation between the focal movement and the postural responses. Therefore, we could expect that postural responses during the focal movement and after its completion would behave differently under conditions of postural threat.
Round 4
Reviewer 3 Report
Comments and Suggestions for Authors
No more comments